# Angiogenesis in the Outer Membrane of Chronic Subdural Hematomas through Thrombin-Cleaved Osteopontin and the Integrin α9 and Integrin β1 Signaling Pathways

**DOI:** 10.3390/biomedicines11051440

**Published:** 2023-05-13

**Authors:** Koji Osuka, Yusuke Ohmichi, Mika Ohmichi, Satoru Honma, Chiharu Suzuki, Masahiro Aoyama, Kenichiro Iwami, Yasuo Watanabe, Shigeru Miyachi

**Affiliations:** 1Department of Neurological Surgery, Aichi Medical University, 1-1 Yazakokarimata, Nagakute 480-1195, Japan; 2Department of Anatomy II, Kanazawa Medical University, 1-1 Daigaku, Uchinada, Kahoku 920-0293, Japan; ohmy@kanazawa-med.ac.jp (Y.O.);; 3High Technology Research Center, Pharmacology, Showa Pharmaceutical University, 3-3165 Higashi-Tamagawa Gakuen, Machida 194-8543, Japan

**Keywords:** angiogenesis, chronic subdural hematoma, integrin α9, integrin β1, N-terminal half of osteopontin

## Abstract

Background: A chronic subdural hematoma (CSDH) is considered to be an inflammatory and angiogenic disease. The CSDH outer membrane, which contains inflammatory cells, plays an important role in CSDH development. Osteopontin (OPN) is an extracellular matrix protein that is cleaved by thrombin, generating the N-terminal half of OPN, which is prominently involved in integrin signal transduction. We explored the expression of the N-terminal half of OPN in CSDH fluid and the expression of integrins α9 and β1 and the downstream components of the angiogenic signaling pathways in the outer membrane of CSDHs. Methods: Twenty samples of CSDH fluid and eight samples of CSDH outer membrane were collected from patients suffering from CSDHs. The concentrations of the N-terminal half of OPN in CSDH fluid samples were measured using ELISA kits. The expression levels of integrins α9 and β1, vinculin, talin-1, focal adhesion kinase (FAK), paxillin, α-actin, Src and β-actin were examined by Western blot analysis. The expression levels of integrins α9 and β1, FAK and paxillin were also examined by immunohistochemistry. We investigated whether CSDH fluid could activate FAK in cultured endothelial cells in vitro. Results: The concentration of the N-terminal half of OPN in CSDH fluid was significantly higher than that in the serum. Western blot analysis confirmed the presence of these molecules. In addition, integrins α9 and β1, FAK and paxillin were localized in the endothelial cells of vessels within the CSDH outer membrane. FAK was significantly phosphorylated immediately after treatment with CSDH fluid. Conclusion: Our data suggest that the N-terminal half of OPN in CSDH fluid promotes neovascularization in endothelial cells through integrins α9 and β1. The N-terminal half of OPN, which is part of the extracellular matrix, plays a critical role in the promotion of CSDHs.

## 1. Introduction

Chronic subdural hematomas (CSDHs) occur in senior citizens who have suffered from mild head trauma. Episodes of mild head trauma are frequently unrecognized for elderly people. Computerized tomography (CT) scanning often reveals CSDHs. A CSDH is a neovascularized inflammatory disease. However, the pathogenesis of CSDHs has not been fully clarified. The dura is lined with a layer of connective tissue cells named “dural border cells”, which can develop into fibro-cellular connective tissue [1]. After mild head trauma, laceration of dural border cells develops into the inner and outer membranes. Fluids and blood are contained between these membranes. The inner membranes consist of collagen and fibroblasts, which are not involved in CSDH growth. On the other hand, the outer membranes contain inflammatory cells such as neutrophils and eosinophils and produce inflammatory and angiogenetic mediators. High concentrations of growth factors and inflammatory mediators in CSDH fluid have been reported and are involved in angiogenesis within the outer membrane. Vascular endothelial growth factor (VEGF) and angiopoietin are examples of these growth factors [2,3], and inflammatory mediators include interleukin-6 and high-mobility group box 1 (HMGB1) [4,5]. These factors in CSDH fluid activate many kinds of signaling pathways in endothelial cells [6,7,8,9], which induces angiogenesis and growth of the CSDH outer membrane.

Osteopontin (OPN) is an extracellular matrix (ECM) protein that was first identified in osteoblasts and has been implicated as an important factor in bone remodeling. OPN plays critical roles in physiological and pathological processes, including inflammation, through integrin receptors. After cerebral ischemia, OPN induces macrophage migration/activation and matrix remodeling, which provides new matrix–cell interactions [10]. Macrophage-derived OPN extends the astrocyte process and repairs blood–brain barrier dysfunction after ischemic stroke [11]. Thrombin-cleaved OPN (the N-terminal half of OPN) plays a more effective protective role than full-length OPN after focal cerebral ischemia in mice [12] and is useful as a blood biomarker of acute atherothrombotic cerebral ischemia [13]. The N-terminal half of OPN is involved in the pathogenesis of diabetic retinopathy, in which retinal vessel growth and angiogenesis play an important role [14]. These previous data suggest that the N-terminal half of OPN is also closely involved in inflammatory central nervous diseases.

Considering the above, we hypothesized that the N-terminal half of OPN is involved in angiogenesis within the outer membranes of CSDHs. Therefore, we explored whether the N-terminal half of OPN exists in CSDH fluid and investigated the expression of integrin and its subsequent signaling pathway molecules in the outer membrane of CSDHs using Western blotting and immunohistological analyses.

## 2. Materials and Methods

### 2.1. Materials

All chemicals were purchased from Sigma Chemical (St. Louis, MO, USA) unless otherwise specified.

### 2.2. Patients

This study included twenty patients who underwent surgical trepanation surgery for CSDH at our Medical University Hospital. These patients, thirteen men and seven women, ranged in age from 41 to 83 years (mean age of 75 years). The Certified Clinical Research Review Board of our Medical University Hospital approved this study (17-H047). Informed consent was obtained from each patient or the patient’s family.

### 2.3. Analysis of Thrombin-Cleaved Osteopontin in CSDH Fluid

Fluids from 20 consecutive CSDHs were obtained during trepanation surgery. As a control, serum samples were obtained from 5 of these patients. After collection, all samples were immediately centrifuged, and the supernatant was stored at −80 °C until analysis. We measured the concentration of the N-terminal half of OPN using enzyme-linked immunosorbent assay (ELISA) kits (IBL, Gunma, Japan) according to the manufacturer’s instructions. The mean minimum detectable dose of these assays was 8.3 pg/mL for the N-terminal half of OPN.

### 2.4. Western Blotting Analysis

Eight samples of the outer membrane of CSDHs obtained from patients during surgery were included in the study. The membranes were homogenized in 75 μL of homogenization buffer containing 50 mM Tris base/HCl (pH 7.5), 0.1 mM dithiothreitol, 0.2 mM ethylenediaminetetraacetate, 0.2 mM ethylene glycol bis(aminoethyl ether)tetraacetate, 0.2 mM phenylmethylsulfonyl fluoride, 1.25 μg/mL pepstatin A, 0.2 μg/mL aprotinin, 1 mM sodium orthovanadate, 50 nM sodium fluoride, 2 mM sodium pyrophosphate and 1% Nonidet P-40. The homogenates were centrifuged at 12,000× *g* for 10 min at 4 °C. The proteins in the supernatants were separated using 7.5% sodium dodecyl sulfate (SDS)-polyacrylamide gel electrophoresis.

The proteins in the gel were transferred to polyvinylidene difluoride membranes and incubated with primary polyclonal antibodies against integrin β1 (#34971, Cell Signaling Technology, Danvers, MA, USA), vinculin (#4650, Cell Signaling Technology), talin-1 (#4021, Cell Signaling Technology), focal adhesion kinase (FAK) phosphorylated at Tyr397 (#129840, Gene Tex, Irvine, CA, USA), paxillin (#12065, Cell Signaling Technology), α-actinin (#6487, Cell Signaling Technology), Src (#2109, Cell Signaling Technology), β-actin (#048k486, Sigma) and with a primary monoclonal antibody against integrin α9 (#MAB4574, R&D Systems, Minneapolis, MN, USA). All antibodies were used at a 1:750 dilution except FAK phosphorylated (p-FAK) at Tyr397 at a 1:5000 dilution and incubated with the membrane overnight at 4 °C. After washing away any unbound antibodies, the membranes were incubated with secondary antibodies conjugated to horseradish peroxidase (Sigma) at a 1:3000 dilution for 30 min at room temperature. The blots were developed with ECL Prime (GE Healthcare, Buckinghamshire, UK). The phosphorylated FAK at Tyr397 immunoblots were stripped and reblotted with primary polyclonal antibody against FAK (#3285, Cell Signaling Technology) at a dilution of 1:750 overnight at 4 °C. The membranes were developed with ECL Prime. Positive controls were RAW264.7 cell lysate (Cell Signaling Technology), rat liver lysate (BD Bioscience, Franklin Lakes, NJ, USA), A431 cell lysate (Santa Cruz Biotechnology, Dallas, TX, USA) and rat brain lysate (BD Bioscience).

### 2.5. Histological Examinations

For the analysis of cellular expression of integrin α9, integrin β1, FAK and paxillin, immunohistochemical staining was performed on samples from three out of 20 patients at room temperature using the avidin-biotinylated peroxidase complex (ABC) technique. To preserve the outer membrane of the CSDH samples, the membranes were incubated in 10 mL of ice-cold 4% paraformaldehyde in 0.1 M phosphate buffer (pH 7.4) for 3 h and then embedded in paraffin.

In this study, 10-μm-thick sections were prepared with a microtome and mounted onto MAS-coated glass slides (Matsunami Glass, Kishiwada, Japan). The sections were deparaffinized in xylene, hydrated through an ethanol gradient, and then fully rehydrated in water. Endogenous peroxidase activity was blocked with 0.3% H_2_O_2_ in 100% methanol for 20 min. All sections for immunostaining were processed for microwave-enhanced antigen retrieval. Slide-mounted sections immersed in 0.01 M sodium citrate buffer (pH 6.0) were placed for 15 min in a 700 W microwave oven at maximum power.

Nonspecific immunoreactivity was blocked by incubation with goat or donkey serum for 30 min, depending on the primary antibody. The samples were treated with primary antibodies against integrin α9 (#MAB4574, R&D Systems) at a dilution of 1:150, integrin β1 (#34971, Cell Signaling Technology) at a dilution of 1:100, paxillin (#125891, Gene Tex, Gene Tex, Irvine, CA, USA) at a dilution of 1:500 and FAK (#3285, Cell Signaling Technology) at a dilution of 1:500 for 36 h at 4 °C. After several rinses in PBS, the samples were incubated with secondary biotinylated antibodies (anti-goat IgG 1:200, anti-rabbit IgG 1:200; Santa Cruz Biotechnology) at room temperature for 2 h. After several more rinses in PBS, the samples were incubated with Vectastain ABC reagent (Vectastain ABC Kit; Vector Laboratories, Burlingame, CA, USA) for 1 h. After several more rinses in PBS, the bound peroxidase was visualized by incubating the sections with a solution containing 0.05% 3,3′-diaminobenzidine tetrahydrochloride (Sigma Aldrich) and 0.01% H_2_O_2_ in 0.05 M Tris-HCl (pH 7.4) for 10 min. After several rinses in water, the immunostained sections were dehydrated and cover-slipped with Entellan new (Merck, Kenilworth, NJ, USA).

### 2.6. Cultured Vascular Endothelial Cells

Endothelial cells of the mouse brain (b. End3) were obtained from the HPA Cultured Collection (London, UK). The endothelial cells were cultured in Dulbecco’s modified Eagle’s medium (Nissui, Tokyo, Japan) with 10% fetal bovine serum at 37 °C and 5% CO_2_.

### 2.7. Effect of CSDH Fluid on FAK and Extracellular Signal-Regulated Kinase (ERK)

We obtained CSDH fluid during trepanation surgery and centrifuged it to remove any debris. b.End3 cells were incubated with a serum-containing medium that contained CSDH fluid. The volumes of the medium and CSDH fluid were 7.5 mL and 2.5 mL per culture dish, respectively. Protein lysates were prepared from the cells harvested at 5 min, 15 min and 60 min (n = 3 per group). We used b.End3 cells treated with media alone as the control (n = 3). Total cell lysates were subjected to Western blotting analysis using antibodies against p-FAK at Tyr397 (#129840, Gene Tex), FAK (#3285, Cell Signaling Technology), p-ERK at Thr202/Tyr204 (#4695, Cell Signaling Technology), ERK (#9102, Cell Signaling Technology) and β-actin as discussed above. All antibodies were used at a 1:750 dilution except p-FAK at Tyr397 at a 1:5000 dilution. The band intensities were quantitated using densitometry with ImageQuant software (GE Healthcare).

### 2.8. Statistical Analysis

Data are expressed as the mean ± standard error. The Mann–Whitney U test was used for the analysis of differences between the two groups. Statistical analyses were performed using a one-way analysis of variance (ANOVA) followed by Fisher’s post hoc test, as appropriate. Significance was indicated when *p* < 0.05.

## 3. Results

### 3.1. Concentration of N-Terminal Half OPN in CSDH Fluid and Serum

First, we examined whether the N-terminal half of OPN exists in CSDH fluid and serum to determine whether the ECM is involved in the development of CSDH. The concentration of the N-terminal half of OPN in CSDH fluid (29,451.5 ± 8146.5 pg/mL) was significantly higher than that in serum (365.8 ± 73.2 pmol/L) based on the Mann–Whitney U test (*p* < 0.01, Figure 1).

### 3.2. Western Blot Analysis of Integrins and the Angiogenic Signaling Pathway

The N-terminal half of OPN binds to the integrin receptor on the cell surface. Signal transduction from the ECM to cytoplasmic actin occurs through integrin receptors. We explored whether these molecules are expressed in the CSDH outer membrane. Figure 2 shows the results of the Western blot analyses of integrins α9 and β1 and the downstream signaling pathway components. Nearly constant β-actin levels were detected in the CSDH outer membrane samples. Integrins α9 and β1, vinculin, talin-1, FAK, paxillin, α-actinin and Src were detected in almost all samples; however, in some cases, the signals were weak. Moreover, the activated form of FAK was detected in the CSDH outer membrane. Positive controls revealed that these molecules had been correctly detected.

### 3.3. Histological Observations

Next, to determine where integrins α9 and β1, FAK and paxillin were expressed in the CSDH outer membrane, we performed an immunohistochemical analysis. The analysis revealed that integrin α9 (Figure 3A,B), integrin β1 (Figure 3C,D), FAK (Figure 3E,F) and paxillin (Figure 3G,H) were localized in the endothelial cells of blood vessels within the outer membrane. Higher magnification images distinctly showed that the endothelial cells expressed these molecules (Figure 3B,D,F,H). The endothelial cells were consistently negative for the markers listed above in the controls without the primary antibodies (Figure 3I). In the adjacent dura mater, there was no apparent staining for these antibodies except for the endothelial cells in tortuous arteries that penetrated the dura mater (Appendix A).

### 3.4. Activation of FAK and ERK in Endothelial Cells by CSDH Fluid

Furthermore, to determine the effect of the N-terminal half of OPN within CSDH fluid, we examined whether CSDH fluid induced phosphorylation of FAK and ERK in endothelial cells (Figure 4A and Figure 4B, respectively). Significantly higher levels of phosphorylated FAK (at Tyr397) and ERK (at Thr202/Tyr204) were achieved 5 min after the addition of CSDH fluid to cultured vascular endothelial cells than in the control (*p* < 0.05), whereas the levels of FAK, ERK and β-actin were not changed.

## 4. Discussion

The expression of the N-terminal half of OPN in CSDH fluid was significantly higher than that in serum. Integrins and their downstream angiogenic pathway intermediates were detected in the outer membrane of CSDHs. Integrins α9 and β1, FAK and paxillin were expressed in the endothelium of blood vessels in the CSDH outer membrane, and CSDH fluid activated FAK in endothelial cells immediately after exposure.

The extracellular matrix protein osteopontin is a glycoprotein involved in physiological and pathological events during inflammatory processes. The concentrations of thrombin-cleaved osteopontin in synovial fluid were well correlated with the severity of knee osteoarthritis [15]. Compared with full-length osteopontin, the N-terminal half of OPN induces markedly more cell attachment through integrin receptors [16]. Disease activity in lupus nephritis is correlated with the urine concentrations of the N-terminal half of OPN rather than full-length osteopontin, suggesting that the N-terminal half of OPN is an indicator of inflammation of the kidney [17]. A previous study revealed that excessive coagulation, generation of thrombin and increased fibrinolysis occur within CSDH fluid [18]. Given the results of previous studies and our data, osteopontin is cleaved by thrombin within CSDH fluid, and the N-terminal half of OPN plays a role in inflammatory reactions in the CSDH outer membrane. To the best of our knowledge, this study is the first to demonstrate the existence of the N-terminal half of OPN in CSDH fluid.

Integrins are α/β heterodimeric cell surface receptors that mediate cell-cell and cell-ECM interactions and orchestrate cell attachment, movement, growth, differentiation and survival. Integrin α9 is widely expressed in a variety of cell types, including the epithelium [19]. Integrin β1 is the main β subunit for α9 in these cells [19]. The β1 class of integrins participates in many aspects of vascular biology, particularly angiogenesis [20]. β1 integrins play an important role in endothelial cell adhesion, migration and survival during angiogenesis and vascular remodeling [21,22]. A deficit of endothelial β1 integrins prohibited endothelial cell maturation, migration and sprouting and induced endothelial cell apoptosis [23]. Thrombin-cleaved osteopontin can attach to integrin α9β1 via the sequence SVVYGLR, which is located between the arginine-glycine-aspartic acid (RGD) sequence and the thrombin cleavage site [24]. Based on our data, it is possible that thrombin-cleaved osteopontin, i.e., the N-terminal half of OPN, induces angiogenesis through integrin α9β1 in the endothelium of the outer membrane. However, it should be noted that CSDH fluid contains matrix metalloproteinases (MMPs), which contribute to inflammatory processes [25,26]. MMP-3 and MMP-7 can also cleave OPN [27], and this cleaved OPN can bind to integrin α9β1 [28], suggesting that other molecules might be involved in the activation of the integrin α9β1 receptor.

After these integrin receptors combine with the extracellular matrix, the formation of complex multiprotein structures occurs. FAK is a regulator of signals from the ECM to the cytoplasmic actin cytoskeleton (Figure 5) [29]. Angiogenesis is mandatory for tumor development. FAK participates in endothelial cell proliferation, which has been revealed to control tumor angiogenesis in many cancers [30]. FAK promotes angiogenesis in a dose-dependent manner [31] when overexpressed in transgenic mice [32]. α-Actin is a highly conserved protein and a member of the actin cross-linking protein family. α-Actin is phosphorylated on tyrosine residues by FAK and binds to actin [33]. These molecules regulate the flow of signals from the extracellular matrix to the actin cytoskeleton and induce angiogenesis (Figure 5).

Paxillin is an adaptor protein located at the interface between the actin cytoskeleton and the plasma membrane [34] and is one of the key components of integrin signaling (Figure 5). The FAK/Src complex phosphorylates ERK and tyrosine and serine residues of paxillin, promotes cell migration and regulates adhesion turnover at the cell front through paxillin [35]. In alkali-burned corneas, neovascularization occurs from the corneal limbus to the cornea, where paxillin induces the migration of endothelial cells and promotes angiogenesis [36]. Netrin-1 is a laminin-like secreted protein that is thought to be an axon guidance molecule during neural development. Netrin-1 activates the FAK/Src/paxillin pathway and modulates angiogenesis, and these changes are accompanied by the upregulation of VEGF [37]. In human retinal angiogenesis, VEGF-induced FAK/Src/paxillin signaling plays an important role [38]. Both talin and vinculin also play important roles in cell growth, morphogenesis, and cell migration during development. Marked defects in focal adhesions and embryonic death occur with the loss of either talin or vinculin in mice [39,40]. Talin is also a key regulator of the interaction between the cytoskeleton and integrins, having multiple interaction sites for other adhesome components (Figure 5) [41]. Talin-1 is essential for endothelial proliferation and postnatal angiogenesis [42]. Furthermore, vinculin is a key regulator of cell adhesion by engaging in direct interactions with talin and actin (Figure 5) [43,44]. After human corneal limbal epithelial cells were exposed to hypoxic conditions, the expression of talin, paxillin and vinculin differed from that under normoxic conditions, affecting cell migration [45]. Our data revealed that all of these molecules were present, as indicated by Western blot analysis, and were located in the endothelium of the CSDH outer membrane, as indicated by immunohistochemistry. Moreover, FAK and ERK in endothelial cells were activated by CSDH fluid. The N-terminal half of OPN is activated by thrombin signals to the actin cytoskeleton via integrin α9β1 located on the cell surface and induces angiogenesis within the CSDH outer membrane.

Angiogenesis is a complex process regulated by numerous receptors, growth factors, ECM-cell interactions, etc. The concentration of VEGF in hematoma fluid has been reported to be an essential mechanistic factor in the pathophysiological progression and development of CSDH and angiogenesis [3,26]. The angiogenic effect of VEGF depends on the presence of integrin β1 [46]. FAK and Src participate in angiogenesis induced by VEGF [47,48]. We previously revealed that activation of mitogen-activated protein kinases (MAPKs) by VEGF occurs in CSDH outer membranes and plays a critical role in the angiogenesis of CSDHs [9,49]. Phosphorylated c-Jun N-terminal kinase (JNK) is expressed in the vascular endothelium of the CSDH outer membrane. FAK activates JNK through an extraordinary mechanism involving the recruitment of paxillin to the plasma membrane [50]. Overall, this work could serve as a basis for further consideration of the N-terminal half of the OPN/integrin pathway as a potential therapeutic target for CSDH.

In the present study, we note several limitations. First, from our limited number of patients, we could not detect correlations among the concentrations of the N-terminal half of OPN in CSDH fluid, the data from Western blot analyses and the development stage of CSDH. Further studies including more patients will be necessary to clarify this relationship. Second, integrins α9 and β1 and subsequent angiogenic signaling molecules were found only in the outer membrane of CSDHs. We need to determine whether these signaling molecules are activated during the development of CSDH. In our in vitro study, we should treat endothelial cells with sera from healthy people and not medium alone as a control treatment.

## 5. Conclusions

For the first time, we detected the expression of the N-terminal half of OPN in CSDH fluid and integrins α9 and β1, FAK, paxillin, vinculin and the subsequent angiogenic signaling pathway in the CSDH outer membrane. Significantly high concentrations of the N-terminal half of OPN in CSDH fluid might play an important role in angiogenesis and inflammation in CSDH, resulting in the growth of the hematoma. This angiogenic signaling pathway through integrins α9 and β1 might be an alternative therapeutic target for the treatment of refractory CSDH.

## Figures and Tables

**Figure 1 biomedicines-11-01440-f001:**
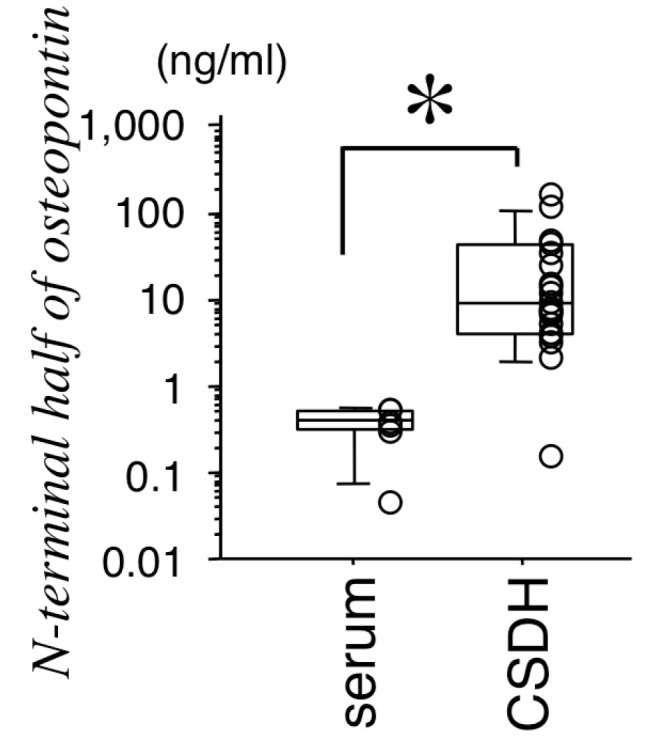
Concentrations of thrombin-cleaved osteopontin (N-terminal half of OPN) in serum (n = 5) and chronic subdural hematoma (CSDH, n = 20). The concentration of the N-terminal half of OPN in CSDH fluid was significantly higher than that in serum based on the Mann–Whitney U test. Data represent the median values and 25th and 75th percentiles with maximum/minimum whiskers with scatter plots. * *p* < 0.01 according to the Mann–Whitney U test.

**Figure 2 biomedicines-11-01440-f002:**
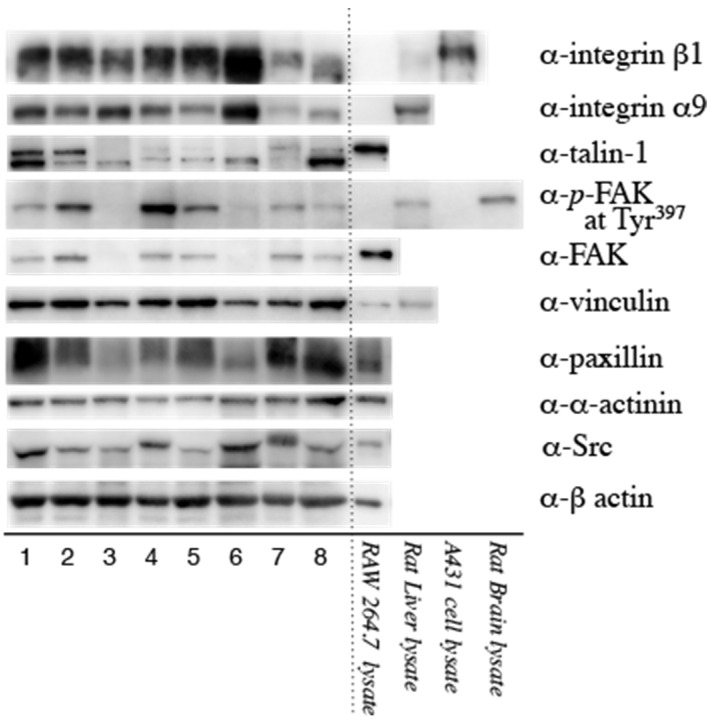
Western blots showing the expression of integrin α9β1 and the subsequent angiogenic pathway molecules in the outer membrane of chronic subdural hematomas from eight patients. Integrins β1 and α9, vinculin, talin-1, focal adhesion kinase (FAK), FAK phosphorylated at Tyr397 (p-FAK at Tyr397), paxillin, α-actinin and Src were detected in almost all cases. Positive controls are shown in the right lanes and suggest that these molecules were correctly detected. RAW 264.7, murine leukemia macrophage cell line lysate; rat liver, rat liver whole cell lysate; A431 cell lysate, epidermoid carcinoma cell lysate; rat brain lysate, rat brain whole cell lysate.

**Figure 3 biomedicines-11-01440-f003:**
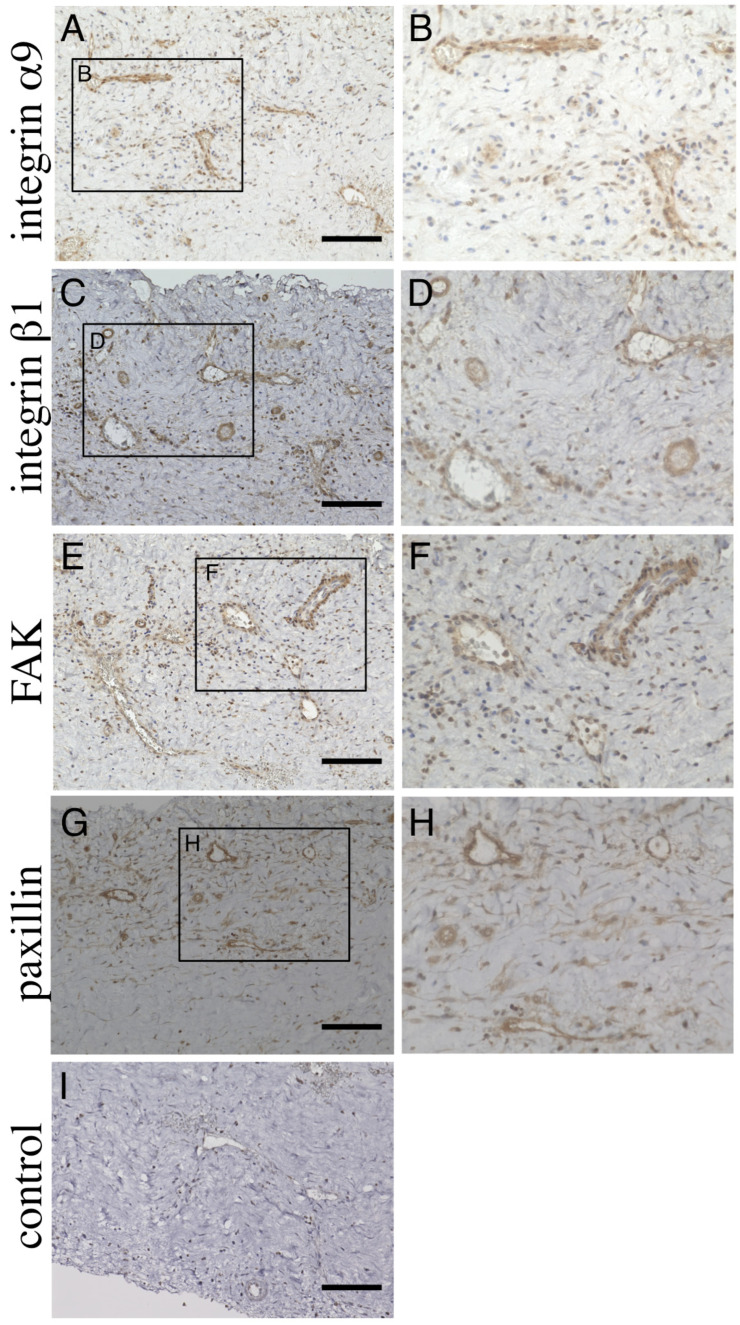
Immunohistochemical analysis of chronic subdural hematoma outer membrane. Ten-micrometer consecutive slices were immunostained with polyclonal antibodies against integrin α9 (**A**,**B**), integrin β1 (**C**,**D**), focal adhesion kinase (FAK, **E**,**F**) and paxillin (**G**,**H**) using the ABC method. The areas within the rectangle, labeled (**A**,**C**,**E**,**G**), are shown at a higher magnification in Panels (**B**,**D**,**F**,**H**), respectively. Note that these molecules are expressed in endothelial cells (**B**,**D**,**F**,**H**). Slices immunostained without primary antibodies are shown in (**I**). Scale bars = 100 µm.

**Figure 4 biomedicines-11-01440-f004:**
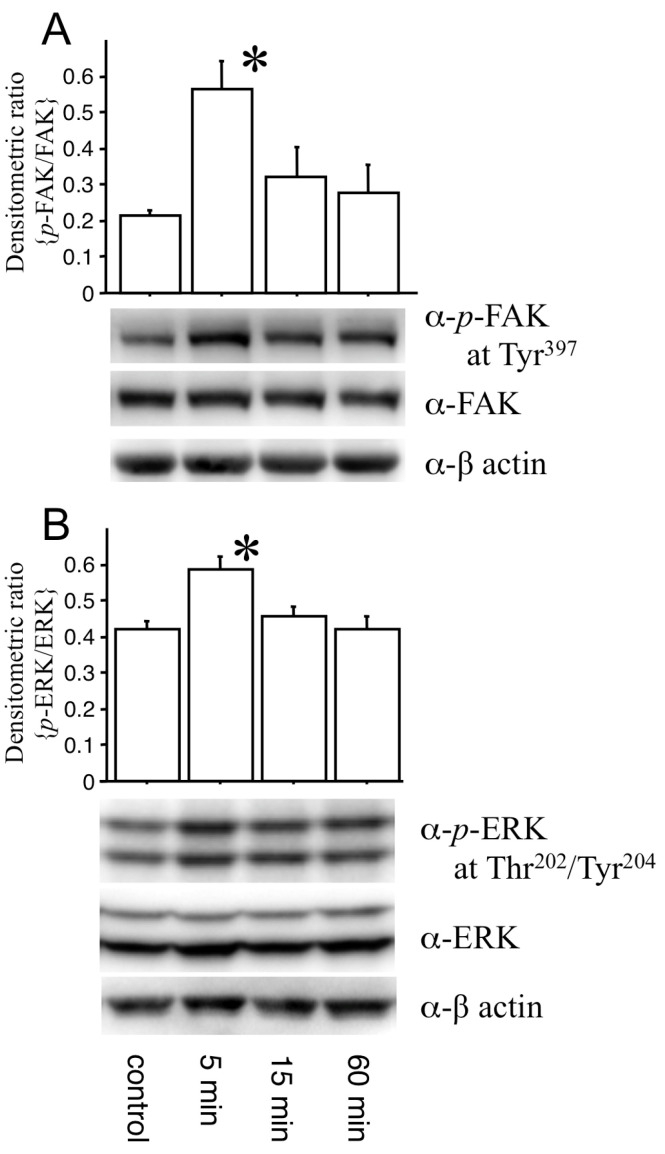
Cultured b.End3 cells were incubated with chronic subdural hematoma fluid for 5, 15 and 60 min. bEnd3 cells treated with media alone were used as the control. Cell lysates were subjected to Western blotting with anti-phosphorylated focal adhesion kinase at Tyr397 (α-p-FAK), anti-FAK (α-FAK), anti-phosphorylated extracellular signal-regulated kinase at Thr202/Tyr204 (α-p-ERK), anti-ERK (α-ERK) and anti-β-actin (α-β-actin) antibodies. The histograms show the amount of p-FAK relative to total FAK and p-ERK relative to total ERK (**A**,**B**, respectively). Phosphorylation of FAK and ERK significantly increased in the cultured b.End3 cells compared with the controls 5 min after treatment. The mean ± SE values from the data of 3 series are shown. * *p* < 0.05 vs. the control (1-way ANOVA followed by Fisher’s PLSD).

**Figure 5 biomedicines-11-01440-f005:**
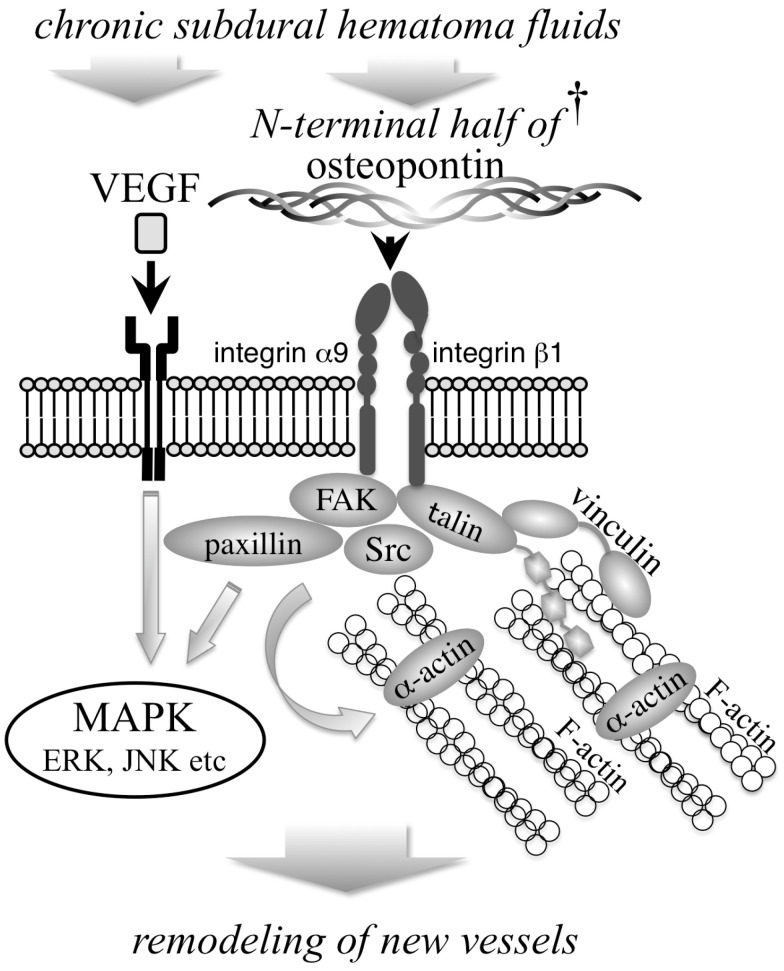
The signal transduction of the N-terminal half of osteopontin (OPN) and VEGF in the outer membrane of chronic subdural hematoma (CSDH). The N-terminal half of OPN binds to the integrin α9β1 receptor on the cell surface. Vinculin, talin, FAK, paxillin and Src are the main regulators that transduce the signal from the integrin located on the cell surface to the actin cytoskeleton. VEGF is also involved in angiogenesis, activating MAPK in the CSDH outer membrane. These pathways collaborate with each other and induce angiogenesis. † This study revealed for the first time that the N-terminal half of OPN is present within CSDH fluid. ERK, extracellular signal-regulated kinase; JNK, c-Jun N-terminal kinase; FAK, focal adhesion kinase; MAPK, mitogen-activated protein kinase; VEGF, vascular endothelial growth factor.

## Data Availability

Please contact the corresponding author.

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
