# Peer review of "Angiogenesis in the Outer Membrane of Chronic Subdural Hematomas through Thrombin-Cleaved Osteopontin and the Integrin α9 and Integrin β1 Signaling Pathways"

_biomedicines, 2023, doi:10.3390/biomedicines11051440_

Round 1

Reviewer 1 Report (Previous Reviewer 1)

The authors did an appropriate revision. The manuscript can be accepted as it is.

The authors did an appropriate revision. The manuscript can be accepted as it is.

Reviewer 2 Report (Previous Reviewer 2)

I think your alterations and additions have enhanced the paper.

Reviewer 3 Report (Previous Reviewer 3)

The authors did an appropriate revision of their manuscript.

Sufficient, no perfect

This manuscript is a resubmission of an earlier submission. The following is a list of the peer review reports and author responses from that submission.

Round 1

Reviewer 1 Report

In the present paper, the authors explored the expression of the N-terminal half of osteopontin (OPN) in chronic subdural hematoma (CSDH) fluid and the expression of integrins α9 and β1 and the downstream components of the angiogenic signaling pathways in the outer membrane of CSDHs. The major finding of this study is that the N-terminal half of OPN in CSDH fluid promotes neovascularization in endothelial cells through integrins α9 and β1 and that the N-terminal half of OPN, which is part of the extracellular matrix, plays a critical role in the promotion of CSDH.

Even though there may be value in the presented results, there are some major issues with the manuscript that need to be addressed if there is a chance for it to be published in Biomedicines journal.

Here I will list some of the major and minor concerns of this study and the article.

1.      In the title it is quite unclear what are the subjects of the study and I feel that this needs to be included. This is also lacking in the abstract, as the patients, as the subject, are not at all mentioned in it.

2.      The introduction seems inadequate in a sense that it is not sufficiently convincing as to why it was important to do this study. It should be rewritten in a way that includes the topic that will be covered, the background of the topic, and info as to why the research matters in the context of the research field, and I do not see that this was fulfilled here.

3.      In chapter 2. Materials and methods, subchapter 2.1. does not have a title.

4.      In subchapter 2.3., it would be useful for the reader to know some more info as to what is the fluid from the CSDHs. In fact, a bit more details on the CSDHs, in general, could have been presented in the introduction (e.g. histology, how it is formed).

5.      At the beginning of subchapter 2.4. the sentence is “Eight samples of the outer membrane of patients with CSDH…” but this does not make sense. The authors, I assume, meant “Eight samples of the outer membrane of CSDHs from patients…” or something similar.

6.      The greatest problem of this study is in the results. In Figure 3, the control microscopy image was that of slices without primary antibodies. For western blot, there is no control at all. For some studies of CSDH, researchers use dura mater from “normal” patients. With just samples from the CSDH patients, there is a lack of reference. Are all these proteins not expressed at all in similar adjacent tissues, we have no way of knowing.

7.      In the in vitro part of the study, for the control, endothelial cells treated with media only were used. What is the reason why maybe those cells were not treated with serum samples, similarly like in the first part of the study where ELISA results from CSDH fluid were compared to serum?

8.      The discussion feels lacking in the sense that I do not feel that, for the most part, the “formula” for writing this part of the article was followed. I liked the part on the integrins because in this part the results of the study were correlated with the previous research and an explanation and theory on the significance of the obtained data were clearly presented.

9.      In the whole paper, authors should re-check their use of the abbreviations, e.g. in some places they still use osteopontin instead of the established abbreviation OPN.

In conclusion of this review, this manuscript could be acceptable for publication upon the correction of the mentioned issues and/or adequate explanation for their choices in the study design and implementation. 

Reviewer 2 Report

To my reading, your description does not make clear whether the five serum specimens that you examined were obtained (and obtained reasonably simultaneously) from five of the 20 consecutive chronic sub dural haematomas from which fluid was collected for examination. Also, were the three subdural membranes studied histologically obtained from members of the 20 consecutive chronic subdural haematoma group?

Can you provide information about the interval between the probable head injury that set in motion the chronic haematoma formation and the time when the specimens for examination were obtained?  Was there evidence of increasing brain dysfunction at the time of surgery? Provision of this information, if possible, would probably help the reader think about the implications of your findings and, in particular, about where the N-terminal portions of osteopontin might be formed, e.g. from surviving haematological cellular components or from damaged brain.

It would have been interesting if you had been able to carry out similar measurements of N-terminal osteopontin in the CSF of your patients

In line 321 of the paper you seem to be suggesting a molecular therapeutic approach in the management of chronic subdural haematoma. Is that your intention?

Reviewer 3 Report

The authors detected a higher expression of the N-terminal half of OPN in 20 CSDH fluid samples than that in serum. Integrins and their downstream angiogenic pathway intermediates were detected in the outer membrane of CSDHs. Integrins α9 and β1, FAK and paxillin were expressed in the endothelium of blood vessels in the CSDH outer membrane, and CSDH fluid activated FAK in endothelial cells immediately after exposure. Given the results of previous studies and the presented data, osteopontin is cleaved by thrombin within CSDH fluid, and the N-terminal half of OPN might potentially play a role in inflammatory reactions in the CSDH outer membrane.

The authors present a well performed and very detailed study on cSDH angiogenesic pathophysiology underlined with WesternBlot Analysis, immunohistochemistry and cultured vascular endothelia cells. This study seems to be  the first to demonstrate the existence of the N-terminal half of OPN in CSDH fluid. There are limitations in the small sample size and the lack of clinical correlation (stage of hematoma, correlation of these factors with clinical features), however, the authors provide an in-depth insight in the pathogenesis of cSDH and angiogenesis.

I recommend to shorten several contents (methodology, results and discussion) where repetitive information is found to create a more concise paper.